# Acute Myocardial Infarction and Risk of Cognitive Impairment and Dementia: A Review

**DOI:** 10.3390/biology12081154

**Published:** 2023-08-21

**Authors:** Elizabeth Hui En Thong, Ethan J. W. Quek, Jing Hong Loo, Choi-Ying Yun, Yao Neng Teo, Yao Hao Teo, Aloysius S. T. Leow, Tony Y. W. Li, Vijay K. Sharma, Benjamin Y. Q. Tan, Leonard L. L. Yeo, Yao Feng Chong, Mark Y. Chan, Ching-Hui Sia

**Affiliations:** 1Internal Medicine Residency, National University Health System, Singapore 119074, Singapore; elizabeth.thong@mohh.com.sg (E.H.E.T.); teoyaohao@gmail.com (Y.H.T.); leowaloysius@hotmail.com (A.S.T.L.); 2Department of Medicine, Yong Loo Lin School of Medicine, National University of Singapore, Singapore 117597, Singapore; ethanquek.jw@gmail.com (E.J.W.Q.); loojinghong@gmail.com (J.H.L.); teoyaoneng@gmail.com (Y.N.T.); vijay_kumar_sharma@nuhs.edu.sg (V.K.S.); benjamin_yq_tan@nuhs.edu.sg (B.Y.Q.T.); leonard_ll_yeo@nuhs.edu.sg (L.L.L.Y.); mark_chan@nuhs.edu.sg (M.Y.C.); 3Department of Cardiology, National University Heart Centre Singapore, Singapore 119074, Singapore; cadencey96@gmail.com (C.-Y.Y.); tony.li@mohh.com.sg (T.Y.W.L.); 4Division of Neurology, Department of Medicine, National University Hospital, Singapore 119074, Singapore; yao_feng_chong@nuhs.edu.sg

**Keywords:** cognitive impairment, dementia, acute myocardial infarction, acute coronary syndrome, coronary artery disease, heart failure, percutaneous coronary intervention, coronary artery bypass surgery

## Abstract

**Simple Summary:**

Cognitive impairment (CI) and dementia are common in patients with heart attacks, and both share common cardiovascular risk factors. In our ageing population, the management and recognition of both becomes increasingly relevant. In this review, we explore the pathophysiology behind this relationship between CI and heart attacks/heart disease. We also discuss risk factors for CI in patients with heart attacks, including the impact of age, sex, and heart failure after the heart attack. We explore how interventions for heart attacks, including percutaneous coronary angiography and bypass surgery, seem to be associated with higher rates of CI, though it is not clear whether this is related to the procedure itself or to the similar underlying risk factors. Finally, we explore how medical management, including most medications prescribed for heart attack patients, can have a positive impact on reducing the risk of CI in patients post-heart attack, though one medication (beta-blocker) may be associated with functional decline in patients with existing CI. The early identification of those with dementia or CI who present with a heart attack is important, as the subsequent tailoring of management strategies can potentially improve outcomes as well as guide prognosis.

**Abstract:**

Cognitive impairment (CI) shares common cardiovascular risk factors with acute myocardial infarction (AMI), and is increasingly prevalent in our ageing population. Whilst AMI is associated with increased rates of CI, CI remains underreported and infrequently identified in patients with AMI. In this review, we discuss the evidence surrounding AMI and its links to dementia and CI, including pathophysiology, risk factors, management and interventions. Vascular dysregulation plays a major role in CI, with atherosclerosis, platelet activation, microinfarcts and perivascular inflammation resulting in neurovascular unit dysfunction, disordered homeostasis and a dysfunctional neurohormonal response. This subsequently affects perfusion pressure, resulting in enlarged periventricular spaces and hippocampal sclerosis. The increased platelet activation seen in coronary artery disease (CAD) can also result in inflammation and amyloid-β protein deposition which is associated with Alzheimer’s Dementia. Post-AMI, reduced blood pressure and reduced left ventricular ejection fraction can cause chronic cerebral hypoperfusion, cerebral infarction and failure of normal circulatory autoregulatory mechanisms. Patients who undergo coronary revascularization (percutaneous coronary intervention or bypass surgery) are at increased risk for post-procedure cognitive impairment, though whether this is related to the intervention itself or underlying cardiovascular risk factors is debated. Mortality rates are higher in dementia patients with AMI, and post-AMI CI is more prevalent in the elderly and in patients with post-AMI heart failure. Medical management (antiplatelet, statin, renin-angiotensin system inhibitors, cardiac rehabilitation) can reduce the risk of post-AMI CI; however, beta-blockers may be associated with functional decline in patients with existing CI. The early identification of those with dementia or CI who present with AMI is important, as subsequent tailoring of management strategies can potentially improve outcomes as well as guide prognosis.

## 1. Introduction

With an ageing population comes increased rates of dementia, cognitive impairment (CI) and acute myocardial infarction (AMI) [1,2]. With CI and dementia resulting in individuals having impaired activities of daily living and loss of higher cortical function, the early recognition and management of modifiable risk factors is crucial [3]. CI is often subclinical and infrequently identified in the patient’s medical history; it is not routinely collected or measured in patients with AMI [4,5]. However, CI shares many cardiovascular risk factors with coronary artery disease (CAD), including smoking, diabetes mellitus, hypertension and metabolic syndrome [6,7,8]. As such, meta-analyses have shown that dementia and CI are associated with CAD, with patients with CAD exhibiting cognitive dysfunction across multiple domains [9,10,11,12,13,14]. There is a significant proportion of patients with AMI who have CI, ranging from 22% in a community-based cohort to 35% in patients planned for surgical bypass [15,16]. Studies have shown that, in older adults presenting with AMI, there is a high prevalence of undiagnosed CI [17]. In this review, we summarise the evidence surrounding AMI and its links to dementia and CI, including pathophysiology, risk factors and related interventions.

## 2. Methods

We performed a search of Pubmed, Scopus, Embase and MEDLINE on 21 April 2023 for the literature including these terms: (cognition OR cognitive deficit OR cognitive decline OR cognitive impairment OR dementia OR memory OR neuroimaging) AND (acute myocardial infarction OR myocardial infarct OR heart attack OR acute coronary syndrome). We included studies enrolling adults with AMI that investigated its association with CI or dementia. The primary outcomes were diagnoses of cognitive impairment or dementia. Additional searches were performed on related topics including: (coronary artery disease OR ischaemic heart disease AND cognitive impairment), interventions for AMI (percutaneous coronary intervention OR percutaneous transluminal coronary angioplasty OR coronary balloon angioplast* OR myocardial revascularization OR coronary artery bypass graft OR surgical revasculari*ation) and post-AMI therapies (antiplatelet OR acetylsalicylic acid OR cyclooxygenase inhibitor, statin OR HMG-CoA reductase inhibitors OR lipid-lowering drug, renin-angiotensin system antagonists OR angiotensin-receptor blocker OR angiotensin converting enzyme inhibitor, beta-blockers, cardiac rehabilitation). Studies were screened based on titles and abstracts, and additional articles were manually identified through a search of the literature review references. All relevant articles underwent a full-text review.

## 3. Results

### 3.1. Pathophysiology

#### 3.1.1. Cognitive Impairment and CAD: Pathophysiology

The potential mechanisms of CAD’s links to CI are complex. Major dementias involve some degree of vascular dysregulation, ranging from 34% after severe stroke to 80% in Alzheimer’s disease (AD) [18,19]. Vascular dysregulation can carry up to a two-fold risk that a neurodegenerative substrate will develop into dementia [18,20,21].

In dementia caused by neurovascular disease, atherosclerosis and the degeneration of small blood vessels can result in clinically silent microinfarcts and dysfunctional white matter pathways in the cerebral cortex (Figure 1). This can result in cerebral hypoperfusion and microvascular ischaemia, enlarged periventricular spaces, cerebral amyloid angiopathy, and hippocampal sclerosis [13,14,22,23,24,25,26,27]. Moreover, left ventricular failure and its resultant decreased cardiac output can lead to hypotension and cerebral hypoperfusion [28]. Others propose that increased platelet activation in patients with CAD may result in perivascular cerebral inflammation, cerebral vasoconstriction, and worsening carotid artery disease [29,30]. Platelet activation can also result in the deposition of amyloid precursor protein and amyloid-β (Aβ) protein in the brain, which can potentiate early onset AD [31,32]. Patient compliance with suggested lifestyle modifications and medication can also be affected by CI. With cardiovascular and some neurodegenerative diseases sharing similar risk factors, underlying vascular dysfunction is a key pathology in both Alzheimer’s and Vascular dementias [28].

The function of the nervous and cardiovascular systems are intertwined in a complex manner [33]. Known as the “heart–brain axis”, distant organs work together through neuronal activity, neurohormonal response and vasculature in order to orchestrate a multisystem response and, in some cases, disease process and dysfunction [34]. Neurological diseases can result in arrhythmias and electrocardiogram changes, and acute central nervous insults such as stroke or haemorrhage can result in myocardial injury [34]. The sympathetic and parasympathetic systems also affect cardiac function [33]. At the centre of this is the neurovascular unit, comprising neurons, pericytes, astrocytes, endothelial cells and perivascular cells (Figure 2) [35,36]. The neurovascular unit regulates cerebral perfusion pressure, blood–brain barrier permeability, immune and molecular trafficking and clearance pathways to maintain normal homeostasis at a cellular level [37,38,39]. Vascular aging can target these sensitive regulatory systems, with atherosclerosis and inflammation resulting in neurovascular unit dysfunction, resulting in disordered homeostasis and subsequent CI and dementia [37,38,40,41].

#### 3.1.2. Cognitive Impairment and AMI: Pathophysiology

Some postulate that AMI itself can result in CI (Figure 1 and Figure 2). One mechanism linking AMI to CI includes reduced left ventricular ejection fraction and reduced blood pressure resulting in chronic cerebral hypoperfusion after AMI [42]. In addition to causing microvascular ischaemia, chronic cerebral hypoperfusion can also result in inflammation and the production of pro-inflammatory cytokines, resulting in neurovascular unit dysfunction, altered metabolism in the brain and resultant and structural changes including white matter lesions and reduced grey matter [43,44,45,46]. Reduced left ventricular ejection fraction is also associated with increased atrial filling pressure with subsequent atrial dilatation and remodelling [42]. With this comes increased rates of atrial fibrillation, leading to intracardiac thrombi formation which can dislodge into the brain [42,47]. As such, AMI is associated with subclinical silent infarcts as well as ischaemic stroke, which are, in turn, also associated with dementia [48]. Cardiogenic shock can also cause also cerebral hypoperfusion, increasing the risk of watershed infarcts and microvascular ischaemia in the brain [29,49,50,51]. Indeed, Sundboll found an increased risk of vascular dementia in patients with atrial fibrillation and heart failure one year post-AMI [42].

However, there is evidence that increased rates of CI are seen even in those with preserved ejection fraction (EF) post-MI [52,53,54,55]. Animal studies found that the high oxidative stress levels during the early remodelling phase of AMI contribute to neuronal loss and failure of normal circulatory autoregulatory mechanisms, thereby contributing to CI [56,57,58,59,60,61,62,63,64]. In addition, subclinical left ventricular dysfunction has been shown to be associated with accelerated brain ageing via the “heart-brain axis”, with systemic inflammation resulting in neurovascular unit dysfunction (Figure 1 and Figure 2) [65,66,67,68].

### 3.2. Cognitive Impairment Post-AMI

There are a limited number of studies that have studied CI post-AMI, with the overall consensus being that AMI is associated with CI. The TRIUMPH (Translational Research Investigating Underlying Disparities in Acute Myocardial Infarction Patients’ Health Status) registry was a multi-centre prospective registry of 772 patients without overt dementia [69]. It found that up to half of older AMI patients (>65 years old) have CI one month post-discharge, and 25% had moderate-severe CI. Patients with moderate-severe CI had a higher 1-year mortality rate and a 2.7 greater hazard of death [69]. The prospective population-based cohort study OXVASC (Oxford Vascular Study) similarly found a high prevalence of CI (49%) at one year post-AMI, and that CI was more prevalent in AMI vs. transient ischaemic attack (TIA) patients, but of a similar incidence compared to CI after minor stroke [70]. The Rotterdam population-based cohort study of 6347 patients also found that patients with unrecognised AMI had increased risk of dementia as well as cerebral small vessel disease [71]. The prospective, observational ICON1 (Improve Cardiovascular Outcomes in high-risk older patients with acute coronary syndrome) study similarly found that older patients with NSTEMI and increased frailty (which included cognition as part of its assessment) were more likely to reach major adverse cardiovascular events [17]. The ICON1 study also found that recurrent MI was independently associated with CI at 1 year, after adjusting for age and sex [17]. This implies that early and aggressive therapy to prevent recurrent MI can potentially reduce risk of CI. This was supported by a large population-based cohort study from Denmark of 314,911 patients that found that MI was associated with a higher risk of vascular dementia, though it did not find any association with other subtypes of dementia [42].

On the other hand, other studies have not found any association with dementia and preceding AMI [72,73,74,75,76]. The Cache County Study, a prospective study of 3264 patients, did not find that prior AMI was associated with higher rates of dementia [77]. Other studies, including the Rochester Epidemiology Project [72], have also not found any association between dementia and AMI [72,76]. A meta-analysis, published in 2017, of 24 studies found that, in prospective cohort studies, patients with coronary heart disease (CHD) have a 45% increased odds of dementia, with AMI being associated with increased odds (odds ratio 1.46, 95% CI 1.16–1.84, *p* = 0.001) [10]. However, separate meta-analyses of cross-sectional and case-control studies did not find a significant effect of AMI on CI. This may be contributed to by the moderate heterogeneity between the studies as well as differences in baseline cardiovascular risk factor profile and follow-up duration. In addition, some cross-sectional studies were excluded from the analysis due to different continuous outcome measures for CI, but these excluded studies did note that patients with CHD were associated with CI [10].

Moderate/severe CI was also associated with increased morbidity at 1 year post-MI, though not reaching statistical significance, with higher rates of rehospitalisation, limitations in activities of daily living, and loss of independence [69]. However, this may be contributed to by the fact that patients with CI had less invasive management for their MI (despite presenting with high Killip class), and significantly lower rates of cardiac rehabilitation [69]. Definitive therapy with invasive management and cardiac rehabilitation results in better outcomes and, as such, the lower rates of both may potentially confound the results. In addition, there may be selection bias in this study, as patients who did not complete the interview for cognitive function (and who were therefore excluded from analysis) were older, of a lower educational level and female. As studies have shown that education level and age are inversely proportional to rate of cognitive decline [78,79], this study’s findings are likely conservative.

Overall, there is a trend towards AMI’s association with CI, especially when taking into account their shared cardiovascular risk factors.

### 3.3. Impact of Cardiovascular Risk Factors on Dementia Subtypes

#### 3.3.1. Cardiovascular Risk Factors and Alzheimer’s Dementia

Alzheimer’s Dementia (AD) is the leading cause of dementia, accounting for more than half of all dementia cases [80,81]. Cardiovascular risk factors are associated with AD, with studies finding that vascular disease and infarcts lower the threshold for the development of AD [21,82,83,84]. This may be due to the effect of vascular disease on inflammation, with inflammation being a key pathological process behind AD [85]. The deposition of β-amyloid protein, neurofibrillary tangles and neurotoxic peptide aggregation in the intra- and extracellular regions of the brain induce the inflammatory pathways that result in the accumulation of prostaglandins, cytokines and other inflammatory mediators that cause neuroinflammation [86,87,88]. This is contributed to by platelet activation, which results in the deposition of the amyloid precursor protein and β-amyloid protein in the brain [31,32]. As such, the presence of these inflammatory mediators in the brain parenchyma and cerebrospinal fluid have been shown to be associated with CI [86,87,89,90]. In addition, an analysis of neuroimaging and biomarkers from the Alzheimer’s Disease Neuroimaging Initiative found that neurovascular dysregulation and abnormal proteins in the vascular system are involved in the early pathology of AD [91]. Hence, reducing the risk of vascular disease and dysregulation has the potential to reduce the incidence of AD.

AMI is associated with higher rates of CI, and cardiovascular risk factors are associated with AD. Whilst animal studies have found that AMI evokes neuroinflammation and subsequent AD, very few human studies have studied the direct effect of AMI on AD [92]. Sundboll’s cohort study of 314,911 patients with AMI and 157,319 matched controls did not find AMI to be significantly associated with increased risk of AD [42]. However, the OXVASC study found that there was a greater prevalence of impaired memory and language versus frontal/executive cognitive domains in AMI compared to stroke/TIA patients [70]. This pattern of CI was more similar to that of memory clinic patients and those with AD [70]. This difference in cognitive domain performance suggests that degenerative brain pathology may be the main driver of CI in AMI patients, and that Alzheimer-type pathology may be a key mediator in CI post-AMI. Similarly, Kivipelto et al.’s prospective population-based study of 1449 patients found that a history of AMI was significantly associated with AD, and previous studies have linked heart failure with Alzheimer-type pathology [93,94,95]. This is likely due to associated cardiovascular risk factors as well as the compounded risk from the apoE epsilon-4 allele [96]. The apoE epsilon-4 allele has been found to be a significant risk factor for hypercholesterolaemia [96] and is, at the same time, independently associated with premature AMI [77,97] (see Section 3.9.2 for more details). That being said, it is important to note the difference in cardiovascular risk factor profile for patients presenting with AMI vs. TIA/stroke. There was more smoking, less atrial fibrillation and less hypertension in AMI patients [70]. Smoking is more of a risk factor for AD and memory impairment, whereas atrial fibrillation and hypertension are more strongly associated with vascular dementia, small vessel disease and stroke [21,98,99,100]. This suggests that the different prevalence of individual cardiovascular risk factors in patients presenting with AMI vs. TIA/stroke patients may also contribute to the subtype of CI that later develops.

#### 3.3.2. Cardiovascular Risk Factors and Vascular Dementia

Vascular dementia (VD) accounts for 20% of all dementias in western countries [80,81]. AMI is associated with a high risk of VD, and is largely due to shared underlying risk factors, including obesity, atherosclerosis, diabetes mellitus, metabolic syndrome, hypertension and age [42,101]. These cardiovascular risk factors create a pro-inflammatory substrate for both neurovascular pathology and cardiovascular pathology, driving the neuroinflammatory response including deposition of inflammatory cytokines in the brain parenchyma, and playing a key role in the pathogenesis of VD [101]. Inflammatory cytokines are also found in high concentrations in the brain parenchyma post-AMI, similar to those seen in VD patients, and remain elevated for 4–8 weeks post-AMI [102,103]. Animal studies have also found that the behaviour of post-AMI mice recapitulates that of human VD [104,105]. Interestingly, treatment with anti-TNF-α agents improved their short-term memory, supporting the neuroinflammatory hypothesis [106].

Whilst atheroslcerosis is the primary driver for AMI, the effect of atheroslcerosis on VD is seen over a longer period of time due to small but cumulative neurovascular infarcts resulting in a gradual decline in cognition [42,107]. In addition, AMI is associated with higher risk of ischaemic and haemorrhagic stroke [48,108]. AMI and resultant atrial fibrillation and regional wall motion abnormalities increases risk of left heart thrombus formation, which can embolise to the brain and cause ischaemic stroke [42]. Studies have found that patients with atrial fibrillation or heart failure post-AMI have a higher rate of VD compared to patients without [42,109].

The use of antiplatelet therapy, especially dual antiplatelet therapy for 1 year post-AMI, is also associated with a higher risk of haemorrhagic stroke and cerebral microbleeds, which can also contribute to the development of dementia [110,111,112]. In addition, post-AMI left ventricular failure, cardiogenic shock and atrial fibrillation can lead to decreased cardiac output, hypotension and cerebral hypoperfusion [28]. In patients with existing hypertension and stenosed arteries, this hypotension potentiates watershed infarctions in vulnerable vascular territories and affects neurovascular reactivity, increasing the risk of cognitive decline [28,29]. This interdependent relationship between the nervous and cardiovascular systems is known as the “heart-brain axis” [82], with patients with heart failure having up to 1.6× more cognitive impairment compared to controls [113,114].

#### 3.3.3. Cardiovascular Risk Factors and Dementia with Lewy Bodies

Dementia with Lewy Bodies (DLB) is the third most common neurocognitive disorder after VD, and is the second most common neurodegenerative disease after Alzheimer’s Dementia [80]. DLB classically features parkinsonism alongside fluctuating cognitive impairment, hallucinations and autonomic dysfunction [115]. Previous studies on cardiovascular risk factors and Parkinson’s disease have mixed results, with some studies showing elevated risk, others showing reduced risk and others showing no relationship [116,117,118]. There are limited data on the relationship between cardiovascular risk factors and DLB. Two retrospective observational studies both found that there was a low prevalence and association of cardiovascular risk factors in patients with DLB [119,120]. An autopsy-confirmed study of neurodegenerative disease found a reduced prevalence of vascular disease in DLB compared to AD [18]. Haugarvoll’s cohort study found that cardiovascular risk factors were not associated with increased risk of dementia in patients with Parkinson’s disease [118]. Overall, the pathology driving DLB seems to be unrelated to cardiovascular risk factors.

#### 3.3.4. Cardiovascular Risk Factors and Frontal-Temporal Dementia

Few studies have studied the relationship between cardiovascular risk factors and frontal-temporal dementia. Frontal-temporal dementia can be sporadic or familial, and accounts for up to 10% of dementia cases in patients < 65 years old [121]. A case control study found that diabetes mellitus may be independently associated with frontal-temporal dementia, with no other significant association with other cardiovascular risk factors [122]. Another study found a lower prevalence of hyperlipidaemia and hypertension in frontal-temporal dementia compared to cognitively intact controls [123], and others suggest that patient with frontal-temporal dementia may have a lower prevalence of cardiovascular disease [124]. Further studies are required to better understand these associations and their implications.

### 3.4. Cognitive Impairment Post-Intervention for AMI

Patients undergoing coronary revascularisation may have increased risk for post-procedure CI, though whether this is secondary to the intervention itself or underlying cardiovascular risk factors needs further clarification.

Some studies suggest that coronary artery bypass grafting (CABG) may be associated with post-operative neurological deficits [42,125,126]. Up to 19% of CABG patients have evidence of CI prior to surgery [127], with 42% of patients having CI at 5 years post-CABG [128]. A Danish population-based cohort study found that the risk for CI was greater in patients who underwent CABG compared to who those who did not go for surgery [42]. Similarly, the Cardiovascular Health Study found that CABG was associated with dementia [126]. This may be attributed to the fact that patients who undergo CABG after an AMI may also experience cerebral hypoperfusion and embolisation during clamping or cannulation of the aorta [57]. On the other hand, the SWEDEHEART (Swedish Web-system for Enhancement and Development of Evidence-based care in Heart disease Evaluated According to Recommended Therapies) registry did not find a difference in rates of dementia between patients who underwent CABG (n = 111,335) and match controls (n = 222,396) [129]. In addition, these findings may be confounded by the fact that the patients recommended for CABG (i.e., triple vessel disease or left main disease) are also more likely to have significant atherosclerosis, including in the cerebral vasculature [42]. This suggests that underlying cardiovascular risk factors may drive the observed CI, rather than from the procedural risk itself. The differences in study outcomes may be also accounted for by residual confounding in observational studies, as well as differences in baseline characteristics including age and cardiovascular risk factors.

There are inconsistent data with regards to percutaneous coronary intervention’s (PCI) relationship with CI. Some studies suggest that potential periprocedural dislodgment of atheroma from the cardiac catheter during coronary angiography can result in micro-emboli [130,131,132]. This can potentially travel to the brain and cause ischaemic and/or subclinical stroke, resulting in CI [130,131,132]. Separately, PCI in patients with CI is associated with poorer outcomes compared to PCI in patients without CI, up to three years post-intervention [128]. CI has also been shown to be an independent cardiac prognostic factor [128]. The observational THORESCI (Tilburg Health Outcomes Registry or Emotional Stress after Coronary Intervention, n = 384) study found that patients who underwent PCI for AMI were at higher risk of depression, fatigue, poor concentration and poor attention levels, in comparison to elective PCI patients [132], which can cause significant functional impairment and distress [133]. Indeed, other studies have found that depression is associated with both CAD and cognition and increases the risk of developing CAD [134,135,136], likely due to its pro-inflammatory state with increased levels of IL-1, IL-6 and TNF-alpha [137,138]. Other proposed mechanisms include decreased medication compliance, poor compliance to recommended healthy lifestyle changes [139], sympathoadrenal system hyperactivity [136], platelet hyperresponsiveness [140] and reduced heart rate variability [141]. Frailty rates are higher in patients with CI 1 year post-PCI, in comparison to patients without CI [128]. This is likely because frailty is often related to multisystem disorders including chronic inflammation and immune activation, malnutrition and comorbid chronic illnesses, which are also often associated with cardiovascular disease [142]. As such, CI is an important cardiac prognostic factor post-PCI in elderly patients.

With regards to rates of CI in PCI vs. CABG, Whitlock et al. analysed data from the community-based, observational Health and Retirement Study and found no significant difference in cognitive outcomes between patients who underwent PCI compared to patients who underwent CABG [143]. A 2009 study also found that rates of CI were greater in the CAD group compared to the general population, regardless of intervention [144]. The high rates of CI may simply be attributed to underlying cardiovascular risk factors rather than procedural risks.

### 3.5. The Impact of Age on CI Post-AMI

Age itself is a significant risk factor for cognitive decline, with CI being present in 1% of 60 year olds, and in 30–60% of 90 year olds [145]. Whether or not this confounds results must be evaluated, and is especially relevant as the majority of AMIs affect adults > 65 years old [146,147].

Age itself may worsen the outcomes post-AMI. Data from the SILVER-AMI study found that patients > 75 years old with CI have a higher risk of mortality and functional decline post-AMI, compared to those without CI [146]. Similarly, most studies on AMI and CI recruited patients who are older—for example, the TRIUMPH study had a mean age of 73.2 ± 6.3 years, and the OXVASC study had a mean age of 68.1 ± 12.4 years [69,70]. Both found that AMI was associated with CI in older patients. However, CI is also seen in younger patients post-AMI. Salzwedel’s prospective observational study of 496 patients < 65 years old found that mild CI was detected in 36.7% of pre-morbidly cognitively intact patients at 3 weeks post-AMI, and that it was associated with reduced treatment compliance, physically demanding work and lower education levels [148]. These factors are also known to be associated with poorer cognitive function [78,149,150]. Whilst there was still a significant association between AMI and CI, the rate of CI was lower in younger patients (36.7% in Salzwedel’s study vs. 49% in OXVASC and 55.6% in TRIUMPH) [69,70,148]. Overall, these studies suggest that post-AMI CI is seen in both younger and older populations, though it is more prevalent in the older age group.

In contrast, the THORESCI observational study found that older AMI patients were less likely to have CI and a poorer quality of life (QOL) in comparison to those of a younger age, suggesting that age is inversely related to CI [132]. This may be contributed to by the fact that AMI in younger patients can have a larger perceived impact on their cognition, especially if they are exposed to highly cognitively demanding activities and work. In contrast, older patients are less likely to be working, and may experience a lesser perceived impact on their cognition in comparison to younger patients [132].

There is currently no randomised controlled trial comparing CI post-AMI in older patients treated with PCI vs. medical therapy. The ongoing SENIOR-RITA (non-ST segment elevation myocardial infarction randomised interventional treatment, ClinicalTrials.gov identifier: NCT03052036) trial, due for completion in 2024, will compare outcomes including CI for invasive therapy vs. medical treatment for older patients with non-ST segment elevation myocardial infarction. This will give us insight into how choice of invasive vs. non-invasive therapy affects CI in the elderly.

### 3.6. Sex Differences in Post-AMI CI

It is well known that there are differences in cardiovascular disease between males and females. Whilst cardiovascular disease is the leading cause of mortality for both sexes, males have a higher cardiovascular disease mortality rate compared to females [151,152]. Males are also suggested to be at greater risk of incident of CI or dementia [11].

The Rotterdam Study, a population-based cohort study of 6347 patients, found that men with unrecognised AMI were associated with an increased risk of dementia as measured through magnetic resonance imaging for white matter lesions and brain infarction [57]. No such association was found between unrecognised AMI and women [57]. Similarly, the Cache County Study found that males with a history of AMI had higher rates of Alzheimer’s dementia and vascular dementia in comparison to females, though this did not reach significance [77]. As such, studies have shown that there is increased risk of stroke and cardiovascular morbidity in men with AMI but not women [71,153]. This may be contributed to by higher rates of neurovascular disease in men compared to women [151,152].

Nonetheless, post-AMI, CI is also seen in women. The Bronx Aging Study found that women aged >75 years with a history of AMI were five times more likely to develop dementia than women with no history of AMI [154]. Aronson’s prospective study of dementia-free elderly people found that AMI was associated with dementia, and that women with a history of AMI were at a five times greater risk of developing dementia than women without a history of AMI [154]. Interestingly, this was not seen in men [154]. The WHIMS (Women’s Health Initiative Memory Study) prospective study of 6455 cognitively intact post-menopausal women found that women with AMI or CAD were at high risk of CI, even after excluding those with incident stroke or TIA [155]. This was related to increased rates of hypertension and diabetes in this population [155]. This is supported by the WHIMS-MRI (Women’s Health Initiative Memory Study-Magnetic Resonance Imaging) study, with patients with hypertension and diabetes having a higher prevalence of abnormal white matter lesions, smaller brain volumes and increased ischaemia [37,38].

With a general paucity of evidence investigating the impact of sex on post-AMI CI, at this juncture it is difficult to make any clear conclusions.

### 3.7. The Impact of Post-AMI Heart Failure on CI

Heart failure (HF) is well known to be associated with CI, and can result in poor self-care, higher rates of hospitalisation and increased mortality [114,156,157,158,159,160]. As such, a meta-analysis of cross-sectional and case-control studies found that HF was associated with a 62% higher risk of CI compared to those without HF [114]. Post-AMI HF and reduced left ventricular ejection fraction is similarly associated with permanent CI post-AMI [156]. Interestingly, both HF with reduced ejection fraction and HF with preserved ejection fraction are associated with CI [161,162], and the severity of AMI is not associated with the degree of resultant CI [56]. This is likely related to the above-mentioned mechanisms (Section 3.1.2) involving the “heart-brain axis”, neurovascular unit dysfunction and subsequent neurohormonal and autoregulatory dysfunction (Figure 1 and Figure 2). Overall, studies suggest that post-AMI HF of any degree is associated with CI.

### 3.8. Effect of Post-AMI CI on Prognosis

The Cooperative Cardiovascular Project of 129,092 patients with AMI found that dementia was associated with higher mortality up to one year post-admission [163]. Similarly, the SILVER-AMI study of 3041 patients > 75 years old with AMI found that mild, moderate and severe CI were all associated with increased risk of readmission, with moderate-severe CI associated with increased risk of death [163].

The higher mortality found in dementia patients with AMI may be attributed to a few reasons. Firstly, patients with dementia received significantly fewer invasive therapies (thrombolysis, angioplasty, bypass surgery) compared to those without dementia, which may result in poorer outcomes [116,117,118,119,120,121,122]. Age is highly correlated with dementia, and preference for medical management in view of reduced life expectancy in the elderly may contribute to this decision [163]. Families and physicians may also perceive that the risk profile for a given invasive therapy may compromise quality of life, especially in the elderly who may have a more adverse risk profile from the invasive therapy, and therefore opt for medical management [163]. Nonetheless, the lower rates of invasive therapies begs the question of whether there is an under-provision of care for this vulnerable group, and whether there is an assumption that this group would not benefit from further care.

Patients with dementia were also less likely to receive angiotensin-converting enzyme inhibitors (ACEi), which may contribute to the higher mortality found in this population [163]. Indeed, ACEi and angiotensin receptor blockers (ARBs) may help to reduce risk of developing AD in patients with mild CI [164,165,166,167,168].

In addition, ACEi is part of the cornerstone of AMI treatment, and has been shown to reduce mortality [169,170]. The reduced use of ACEi in patients with dementia may be contributed to by the fact that those with dementia are more likely to be elderly, with the elderly also having lower glomerular filtration rates for given levels of serum creatinine, and may therefore not be suitable for ACEi [171,172]. However, experimental models show increased angiotensin-converting enzyme levels in the temporal cortex of patients with dementia, suggesting that its inhibition may be beneficial in reducing the neurodegenerative pathology that results in dementia [173]. The Syst-Eur (Systolic Hypertension in Europe Placebo-Controlled Trial) trial also found that ACEi and calcium channel blockers reduced rates of dementia [174], and the PROGRESS (Perindopril Protection Against Recurrent Stroke Study) study found reduced rates of vascular events in elderly patients on ACEi [175]. As such, ACEi should be considered in the elderly population, with its benefits extending past those of post-AMI mortality.

Interestingly, patients reported similar QOL post-MI regardless of baseline cognitive status. This may be due to a different interpretation of what constitutes expected QOL, as studies have found that expected QOL differs with cognitive impairment. The assessment of QOL needs to be adjusted for those with CI [176].

### 3.9. Neuroprotective Strategies Post-AMI

#### 3.9.1. Antiplatelet Therapy

Aspirin is commonly used for cardiovascular disease prevention and treatment [177]. It is a non-competitive, irreversible inhibitor of cyclooxygenase (COX) enzymes. It primarily acts to inhibit COX-1 and therefore thromboxane-A2 synthesis at low doses (75–300 mg/day) [178]. This results in decreased platelet aggregation and vasoconstriction, reducing the risk of intravascular thrombus formation [177,178]. It also reduces pro-inflammatory cytokine release mediated by β-amyloid proteins [179,180]. There is an increased expression of COX-1 in neurodegenerative diseases, and aspirin plays a central role in the neuroinflammatory response [181,182,183]. Studies have suggested that patients with dementia are less likely to be prescribed antiplatelet therapy post-AMI in comparison to dementia-free patients, potentially adversely affecting outcomes [184]. This may be due to side effects, increased rate of contraindications amongst the elderly who are also more likely to have dementia, or concerns about polypharmacy [184].

A meta-analysis published in May 2022 found that subjects with pre-existing CAD who took low-dose aspirin had significantly lower rates of AD and VD compared to those who did not [185,186,187]. However, the use of antiplatelet therapy on cognition has not been studied specifically in the post-AMI population, likely due to the key role of antiplatelet therapy in AMI management. Antiplatelet use in dementia and stroke has been well studied, with mixed results. The ASPREE (Aspirin in Reducing Events in the Elderly) and ASCEND (A Study of Cardiovascular Events in Diabetes) randomised controlled trials did not find that aspirin reduced rates of dementia, whilst also increasing risk of bleeding [187,188]. Recent systematic reviews, published in 2020 and 2021, concur with these findings [189,190]. A large number of cohort, observational and case-control studies suggest that aspirin reduces the risk of AD, though further meta-analyses did not find any significant effect of aspirin use on incident AD [191,192,193,194,195,196,197,198,199,200,201]. There are limited placebo-controlled trials regarding antiplatelet use in VD, as the diagnosis of VD is based on a history of cardiovascular risk factors, stroke or AMI, with patients likely to already be on an antiplatelet. One randomised controlled trial, multiple cohort studies and one systematic review have all concurred that antiplatelet therapy may reduce the risk of VD [185,187,202]. A longitudinal study of patients on the South London Stroke Register found that dual antiplatelet therapy (DAPT) significantly reduced rates of cognitive impairment in stroke patients with AF, and another randomised controlled trials reported reduced rates of early neurological deterioration in post-stroke patients on DAPT compared to aspirin alone [200,203]. Differences in study findings may be contributed to by the variability in the definition of CI and variations in trial design.

Overall, the limited evidence in the post-AMI population suggests that antiplatelets may be associated with reduced CI post-AMI, and this is supported by studies on antiplatelets’ effects on CI in other situations.

#### 3.9.2. Statins

Statins play a key in role in preventing ischaemic stroke associated with atrial fibrillation. AMI increases risk of atrial fibrillation via inflammation and atrial diastolic overload, with fast atrial fibrillation increasing oxygen demand and potentiating ischaemia [204,205]. As such, atrial fibrillation is associated with poor clinical outcomes, including risk of mortality, bleeding and hospitalisation from heart failure [205,206,207,208,209]. Wankowicz et al. performed a retrospective multicentre analysis of 2309 patients and found that the addition of a statin to oral anticoagulants helps in the primary prevention of atrial fibrillation-related stroke [208]. Similarly, Choi et al. found that, in patients with ischaemic stroke and atrial fibrillation, high-intensity statins reduced the rate of adverse clinical and cerebral events [209]. Statin use in patients with atrial fibrillation post-AMI may therefore reduce stroke risk and any subsequent CI post-stroke [210].

This is due to statins’ effects in increasing cerebral collateral circulation, which becomes vital in preserving blood supply to the distal circulation when a large artery is occluded during a stroke [211,212]. Statins additionally reduce mortality in atrial-fibrillation-related stroke via its “pleiotrophic” mechanisms on the circulatory system [213]. This includes improving endothelial function, improving ventricular function via the inhibition of myocardial remodelling, prevention of ventricular arrythmias, inhibition of the inflammatory response by reducing levels of interleukin-6 and C-reactive protein, stabilising the plaque and increasing nitric oxide bioavailability [214,215,216,217,218,219]. Statins’ beneficial effects on the circulatory system are hence protective for atrial-fibrillation-related stroke and post-stroke CI.

In addition, elevated serum cholesterol has been shown to be a risk factor for AD. ApoE epsilon-4 has been found to be a significant risk factor for hypercholesterolaemia [95] and is at the same time independently associated with premature AMI [77,97]. It is also is a prevalent genetic risk factor of AD, being expressed in more than half of AD patients, and this may explain why patients with the ApoE epsilon-4 allele are at increased risk of both AMI and AD [220]. Moreover, altered cholesterol homeostasis and CYP46 polymorphisms can result in increased cholesterol deposition in the brain [219]. It has also been linked to increased β-amyloid protein deposition in the temporal lobe, therefore increasing risk of late-onset AD [219]. The transcriptor factor subfamily of Liver X receptors (LXRs) has also been noted to affect lipid gene regulation and the inflammatory response through affecting the production of nitric oxide synthase, COX-1, IL-1 and IL-6 [221]. As such, LXRs polymorphisms may affect the development of AD pathology in animal models [222].

As such, studies have found that statin usage reduces rates of AD up to 40–70% [223,224,225]. However, other studies did not find any role of statins in AD prevention, including CRISP (Cholesterol Reduction in Seniors Program) [226], PROSPER (PROspective study of pravastatin in the elderly at risk) [227], the Cache County study [228] and the Honolulu study [229]. This may be attributed to the more advanced age of participants in this study, with most of these studies being carried out in populations of >65 years old; others have found that hypercholesterolaemia is strongly associated with amyloid deposition in the brain at younger ages (40–55 years old) [230]. Differences in these findings may therefore be linked to hypercholesterolaemia being an early risk factor for AD, and thereby needing statin usage at a younger age to have an effect on risk of developing AD [230].

Recently, a systematic review and meta-analysis of 36 studies published in 2021 found that statin use was associated with a reduced risk of dementia [231]. Individual types and doses of statin had comparable results, and there was no sex difference in the risk reduction profile [231]. In addition, statin usage was not associated with any neurocognitive risk, including in that of older adults [231]. Notably, the studies included were observational studies, which carry an intrinsic risk of bias. There are currently two ongoing double-blind, randomised, placebo-controlled trials, STAREE (STAtins in Reducing Events in the Elderly) and PREVENTABLE (Pragmatic evaluation of events and benefits of lipid lowering in older adults), which will investigate the preventative effects of statin on CI as well as cardiovascular events [232,233].

#### 3.9.3. Renin-Angiotensin System Inhibitors

Renin-angiotensin system (RAS)-related enzymes and peptides are found in the CNS, and are also involved in learning and memory [234]. The angiotensin hypothesis suggests that the RAS may be altered in AD, and contribute to neuronal and cognitive function, with its alteration contributing to progressive deficiencies in cognition [235]. The renin-angiotensin system’s modulatory role in atherosclerosis and vascular inflammation can also contribute to endothelial dysfunction, thereby enhancing the atherogenic process, with its hyperfunction in animal models also contributing to AD [236,237]. Studies have found that angiotensin-converting enzyme (ACE) activity is increased in AD, with ACE activity being proportional to the concentration of Aβ protein found in the brain [238]. This may promote brain inflammation in the pyramidal neurons in the cortex [37,38]. As such, the National Institute on Aging-Alzheimer’s Association guidelines include antihypertensives as part of pre-clinical AD management [239].

RAS’ effect on the CNS is likely due to a few mechanisms. Firstly, centrally acting ACEi (such as ramipril, captopril, perindopril, lisinopril) are lipid soluble and cross the blood–brain barrier [240]. These can potentially act via anti-inflammatory mechanisms directly on the brain, thereby reducing the chronic inflammatory pathways in AD [241]. In addition, centrally acting RAS regulates cerebral blood flow independently of peripheral RAS. Angiotensin II activation results in Type 1 receptor activation (causing vasoconstriction, endothelial dysfunction and vascular remodelling), and Type 2 activation (causing reduced inflammation, vasodilation and neuronal regeneration) [242]. As such, ARBs inhibits Type 1 receptors, whereas centrally acting ACEi decrease the activation of both receptors [242].

Studies have found that ACEi and angiotensin receptor blockers (ARBs) may be associated with lower risk of AD in patients with mild CI [This was also associated with fewer neurofibrillary tangles in the brain [243]. Lehmann et al. performed a meta-analysis of 6037 cases of AD and 12,099 controls, and found that ACE polymorphism can even be considered a risk factor for developing AD [244]. Ohrui’s randomised, prospective, parallel group trial of 183 patients similarly found that ACE inhibitors reduced the rate of cognitive decline in patients with AD. However, others have found that ARBs, but not ACEi, reduce cognitive decline in AD patients [245].

Overall, Ye et al.’s meta-analysis of 12 studies and 896,410 patients found that ACEi and ARBs were associated with a reduced rate of AD [246]. However, further analysis of the three randomised controlled trials included in the analysis did not produce the same result, and did not find that ACEi or ARBs were associated with a reduced rate of AD [246]. This may be contributed to by the small sample size (mean size of 76 patients) and short duration of follow up (mean of 1.2 years). In addition, some other studies, including the ONTARGET (Ongoing Telmisartan Alone and in Combination with Ramipril Global Endpoint Trial, n = 25,620) and TRANSCEND (Telmisartan Randomised Assessment Study in ACE Intolerant Subjects with CVD, n = 5926) parallel trials, did not find that RAS blockade affected cognition [247]. More recently, a 2022 review by Gouveia concurs with an earlier systematic review and meta-analysis from 2020 that RAS inhibition reduces the risk of dementia [248,249].

Whilst there is mixed evidence on RAS inhibition and its effect on CI, the compliance and use of RAS inhibitors as part of post-AMI treatment should nonetheless be strongly encouraged, as alongside its cardiac benefit it may also reduce the risk of developing CI.

#### 3.9.4. Beta-Blockers

Beta-blockers (BB) are often used in neuropsychiatric conditions, including anxiety, essential tremor and migraine prophylaxis [250]. Some studies have found that BB are associated with an increased risk of depression [251,252], others have found the opposite [253] and yet others have found it to be associated with increased rates of lethargy and fatigue [250]. However, there are limited studies investigating BB and its impact on CI.

A study of 357,030 patients hospitalised with heart failure with reduced ejection fraction (HFrEF) found that the benefit of BB in reducing mortality in HFrEF is at least maintained in patients with dementia [254]. Another prospective cohort study also did not find any association between BB use and delirium [255]. In animal models, BB have been found to impair learning and memory through the exacerbation of inflammation and synaptic degeneration [256,257]. A longitudinal study of 18,063 patients found that BB was associated with increased risk of vascular dementia [258]. However, another cross-sectional study of 8279 patients did not find that BB was associated with CI [259], whereas a third study has suggested that BB reduce rates of functional decline in patients with AD through its effects on Aβ production and inflammation [260].

In the AMI population, only two studies have investigated the role of BB in CI. A cohort study of 15,720 patients found weak evidence that BB was also associated with functional decline in patients with existing CI, but not in those with preserved cognitive function [261,262]. Another observational study did not find that BB use affected mortality [263]. This may suggest that BB has a potential role to accelerate functional decline in AMI patients with CI, though further studies are required to investigate this further.

Overall, with such varied evidence, the role of BB in patients with CI is yet to be clearly established. Until such a time, its positive effect on mortality and morbidity in the general AMI population is likely to take precedence.

#### 3.9.5. Cardiac Rehabilitation

Rates of cardiac rehabilitation were interestingly found to be lower in those with CI compared to those with normal cognition [69]. Cardiac rehabilitation improves physical function and promotes healthy lifestyle changes [264], resulting in reduced rates of mortality and major adverse cardiac and neurovascular events [265,266]. Exercise itself also reduces the risk of CI [267]. The mechanisms proposed include effects on neuroplasticity, through cellular and molecular changes that increase grey matter volume in the frontal and hippocampal regions [268], increased levels of neurotrophic factor [269] and increased blood flow [270]. Cardiac rehabilitation is associated with reduced risk factors including hypertension, hyperlipidaemia, endothelial dysfunction and inflammatory markers that are in turn associated with improved cognitive function [271]. As such, there are lower rates of CI in patients with cardiovascular disease who participate in cardiac rehabilitation [272]. More should be carried out to encourage cardiac rehabilitation in patients with CI to reduce further cognitive decline.

## 4. Limitations

There are some limitations to our data. Firstly, there is variability in the definition of CI and its impact on QOL. Some studies used objective measures of CI whereas others used subjective or self-reported measures of CI. Secondly, whilst most studies have found that advanced age is associated with greater risk of CI, one study found that this group also experienced less perceived cognitive decline, in comparison to younger patients of working age [132]. Patients with existing dementia or CI may interpret what constitutes ‘independent’ activities of daily living very differently from a patient without CI [133]. This is relevant as the perception of expected cognition, and therefore its deviation from the baseline, can make it difficult to objectively compare cognition across studies. Thirdly, the cognitive assessment tools used in the studies also varied. The TRIUMPH and OXVASC studies used TICS-m (the Telephone interview for cognitive status-modified), whereas other studies used MoCA (Montreal Cognitive Assessment) to assess cognitive function. Telephone interviews are more limited in their ability to assess complex language tasks and visuo-executive skills, in comparison to face-to-face assessments, and may not pick up single-domain CI and subtle changes in cognition [273]. Fourthly, the time of baseline and follow-up cognitive assessments varies between studies, ranging from 14 days to over 1 year. CI can take years to develop, and the differing duration of study follow up may limit their comparison.

Moreover, observational data show that hospitalised patients with CI are at greater risk of delirium, increased length of stay and increased mortality [274,275]. Delirium itself can also result in significant CI post-intervention, and can last for months [276]. Whilst most studies attempted to exclude patients who were at risk of delirium (e.g., active infection, cardiac arrest, ventricular arrythmia), studies with a shorter follow-up period may have their results confounded by undiagnosed delirium. In addition, patients predisposed to delirium share similar risk factors to CI, including age, pre-existing CI and cardiovascular disease [277]. The exclusion of these patients may also exclude a significant proportion of patients for whom these studies may have relevance. Finally, some studies did not offer invasive therapy to patients who were older, more frail or who had significant CI. As such, the effect of invasive therapy on CI in this population is understudied.

## 5. Implications

Age itself is associated with poorer outcomes in AMI, and CI itself may increase morbidity and mortality post-MI [52,205,206]. Patients with CI are often excluded from clinical trials for cardiovascular medicine, making the evidence for guideline-directed medical therapy in this group less robust. This review helps us to further our understanding and ability to prognosticate outcomes after AMI based on CI. With dementia being an increasing burden on health and social care, risk factor modification and routine cognitive screening may also help the clinician to identify cognitively impaired patients with AMI who are at risk of poorer outcomes, and who may therefore benefit from closer follow-up and more aggressive therapies to preserve their QOL, functional abilities and save on healthcare expenditure. At the same time, there are higher potential risks of invasive treatment in this group, including periprocedural bleeding and drug-drug interactions [278]. We must balance the potential benefits against the treatment risks and goals of care in this group, with older patients with CI possibly prioritising time at home and QOL, over definitive therapy.

## 6. Conclusions

In conclusion, AMI and CI share common risk factors, and AMI itself is associated with increased risk of CI. Patients with CI and acute myocardial infarction (AMI) are noted to have worse outcomes compared to those without dementia, especially in the older age group. Post-AMI HF is associated with CI. Whether sex and invasive therapies affect CI post-AMI has not been clearly established and should be investigated further. Existing medical therapy for AMI is likely to help reduce rates of post-AMI CI as well, though BB can potentially precipitate functional decline. The early identification of those with dementia or CI who present with AMI is important, as the early identification and subsequent tailoring of management strategies can potentially improve outcomes as well as guide prognosis.

## Figures and Tables

**Figure 1 biology-12-01154-f001:**
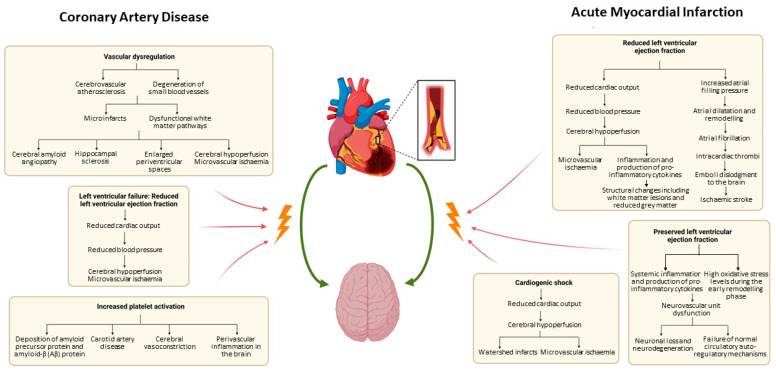
Pathophysiology of coronary artery disease and acute myocardial infarction on the heart–brain relationship.

**Figure 2 biology-12-01154-f002:**
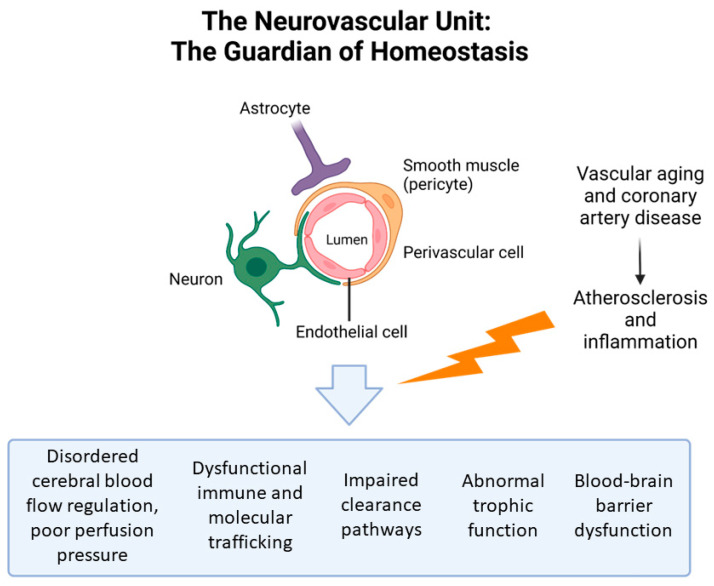
The effect of vascular aging on neurovascular unit and its function.

## Data Availability

No new data were created or analysed in this study. Data sharing is not applicable to this article.

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
