# Peer review of "Acute Myocardial Infarction and Risk of Cognitive Impairment and Dementia: A Review"

_biology, 2023, doi:10.3390/biology12081154_

Round 1

Reviewer 1 Report

In this review, the authors discuss the evidence surrounding AMI and its links to dementia and CI, including pathophysiology, risk factors and related interventions. Vascular dysregulation plays a major role in CI, with atherosclerosis, platelet activation, microinfarcts and perivascular inflammation resulting in neurovascular unit dysfunction, disordered homeostasis and a dysfunctional neurohormonal response. This subsequently affects perfusion pressure, resulting in enlarged periventricular spaces and hippocampal sclerosis. The increased platelet activation seen in coronary artery disease (CAD) can also result in inflammation and amyloid-β protein deposition which is associated with Alzheimer's Dementia. Post AMI, reduced blood pressure and reduced left ventricular ejection fraction can cause chronic cerebral hypoperfusion, cerebral infarction, and failure of normal circulatory autoregulatory mechanisms. Patients who undergo coronary revascularization (percutaneous coronary intervention or bypass surgery) are at increased risk for post-procedure cognitive impairment, though whether this is related to the intervention itself or underlying cardiovascular risk factors is debated. Mortality rates are higher in dementia patients with AMI, and post-AMI CI is more prevalent in the elderly and in patients with post-AMI heart failure. Medical management (antiplatelet, statin, renin-angiotensin system inhibitors, cardiac rehabilitation) can reduce the risk of post-AMI CI; however, beta-blockers may be associated with functional decline in patients with existing CI. In the ending of this review the authors proclame that early identification of those with dementia or CI who present with AMI is important, as subsequent tailoring of management strategies can potentially improve outcomes as well as guide prognosis.

This review is well written and brings new knowledge to the topic of cardiovascular disease.

Nevertheless, in the introduction or discussion section, it is worth referring to current publications describing the significant importance of statins in the prevention of ischemic stroke associated with atrial fibrillation (Wankowicz et al, Choi et al)

Author Response

We thank the editor and reviewers for their comments that help to strengthen the manuscript. We agree that the effect of statins in the prevention of ischemic stroke associated with atrial fibrillation is an important point to be made in the manuscript, and have edited the manuscript to include this under section 3.9.2 ‘Statins’. Once again, we thank the editor and reviewer for the opportunity to improve the quality of the manuscript.

Comment

In this review, the authors discuss the evidence surrounding AMI and its links to dementia and CI, including pathophysiology, risk factors and related interventions. Vascular dysregulation plays a major role in CI, with atherosclerosis, platelet activation, microinfarcts and perivascular inflammation resulting in neurovascular unit dysfunction, disordered homeostasis and a dysfunctional neurohormonal response. This subsequently affects perfusion pressure, resulting in enlarged periventricular spaces and hippocampal sclerosis. The increased platelet activation seen in coronary artery disease (CAD) can also result in inflammation and amyloid-β protein deposition which is associated with Alzheimer's Dementia. Post AMI, reduced blood pressure and reduced left ventricular ejection fraction can cause chronic cerebral hypoperfusion, cerebral infarction, and failure of normal circulatory autoregulatory mechanisms. Patients who undergo coronary revascularization (percutaneous coronary intervention or bypass surgery) are at increased risk for post-procedure cognitive impairment, though whether this is related to the intervention itself or underlying cardiovascular risk factors is debated. Mortality rates are higher in dementia patients with AMI, and post-AMI CI is more prevalent in the elderly and in patients with post-AMI heart failure. Medical management (antiplatelet, statin, renin-angiotensin system inhibitors, cardiac rehabilitation) can reduce the risk of post-AMI CI; however, beta-blockers may be associated with functional decline in patients with existing CI. In the ending of this review the authors proclame that early identification of those with dementia or CI who present with AMI is important, as subsequent tailoring of management strategies can potentially improve outcomes as well as guide prognosis.

This review is well written and brings new knowledge to the topic of cardiovascular disease.

Nevertheless, in the introduction or discussion section, it is worth referring to current publications describing the significant importance of statins in the prevention of ischemic stroke associated with atrial fibrillation (Wankowicz et al, Choi et al)

Response

We thank the editor and reviewers for their comments that help to strengthen the manuscript. We agree that the effect of statins in the prevention of ischemic stroke associated with atrial fibrillation is an important point to be made in the manuscript, and have edited the manuscript to include this under section 3.9.2 ‘Statins’. Once again, we thank the editor and reviewer for the opportunity to improve the quality of the manuscript.

Thank you very much.

Yours Sincerely,

Elizabeth Thong, MBBS (UK), MRCP (UK), MMed (Internal Med)

Department of Cardiology

National University Heart Centre Singapore

1E Kent Ridge Road, NUHS Tower Block Level 9, Singapore 119228

Tel: +65 6779 5555        

E-mail: Elizabeth.thong@mohh.com.sg

Reviewer 2 Report

This is an interesting manuscript on the role of AMI and CAD as risk factors for dementia. However dementia is the result of a highly heterogeneic group of diseases (dementia with Levi`s bodies, AD, PD, frontotemporal dementia etc). I think that the review must be reoorganized. I recommend adding a section to describe at least specific roles of CVDs in Alzheimer dementia, since Alzheimer`s disease accounts for about 70% of all dementia cases worldwide.

Additional comments:

1. What is the main question addressed by the research? Does CAD and AMI contribute to dementia?
2. Do you consider the topic original or relevant in the field? Does it
address a specific gap in the field? Yes, this is important to know risk factors for dementia, especially those related to CVDs.
3. What does it add to the subject area compared with other published
material? This MS gives us an interesting point of view and includes summarized knowledge on the impact of some CVDs in dementia etiology.
4. What specific improvements should the authors consider regarding the
methodology? What further controls should be considered? I recommend adding a section to describe at least specific roles of AMI and CAD in Alzheimer dementia, since Alzheimer`s disease accounts for about 70% of all dementia cases worldwide.   5. Are the conclusions consistent with the evidence and arguments presented
and do they address the main question posed? Yes

Author Response

We thank the editor and reviewers for their comments that help to strengthen the manuscript. We agree that the pathophysiology of dementia varies greatly depending on its subtype. However, the majority of studies looked at the effect of AMI on cognitive impairment or on dementia as a whole, rather than the effects of AMI on the individual subtypes of dementia. At the same time, it is important to acknowledge that how we interpret AMI’s effects on CI and dementia does depend on the subtype of dementia. As such, we have included a section on the impact of cardiovascular risk factors on different dementia subtypes (section 3.3), as this has been studied before. In this section, we explain how cardiovascular risk factors (and thereby indirectly AMI) may affect different dementia subtypes. Once again, we thank the editor and reviewer for the opportunity to improve the quality of the manuscript.

Comment

This is an interesting manuscript on the role of AMI and CAD as risk factors for dementia. However dementia is the result of a highly heterogeneic group of diseases (dementia with Levi`s bodies, AD, PD, frontotemporal dementia etc). I think that the review must be reoorganized. I recommend adding a section to describe at least specific roles of CVDs in Alzheimer dementia, since Alzheimer`s disease accounts for about 70% of all dementia cases worldwide.

Additional comments:

  1. What is the main question addressed by the research? Does CAD and AMI contribute to dementia?
  2. Do you consider the topic original or relevant in the field? Does it address a specific gap in the field? Yes, this is important to know risk factors for dementia, especially those related to CVDs.
  1. What does it add to the subject area compared with other published material? This MS gives us an interesting point of view and includes summarized knowledge on the impact of some CVDs in dementia etiology.
  1. What specific improvements should the authors consider regarding the methodology? What further controls should be considered? I recommend adding a section to describe at least specific roles of AMI and CAD in Alzheimer dementia, since Alzheimer`s disease accounts for about 70% of all dementia cases worldwide.  
  1. Are the conclusions consistent with the evidence and arguments presented and do they address the main question posed? Yes

Response

We thank the editor and reviewers for their comments that help to strengthen the manuscript. We agree that the pathophysiology of dementia varies greatly depending on its subtype. However, the majority of studies looked at the effect of AMI on cognitive impairment or on dementia as a whole, rather than the effects of AMI on the individual subtypes of dementia. At the same time, it is important to acknowledge that how we interpret AMI’s effects on CI and dementia does depend on the subtype of dementia. As such, we have included a section on the impact of cardiovascular risk factors on different dementia subtypes (section 3.3), as this has been studied before. In this section, we explain how cardiovascular risk factors (and thereby indirectly AMI) may affect different dementia subtypes. Once again, we thank the editor and reviewer for the opportunity to improve the quality of the manuscript.

Thank you very much.

Yours Sincerely,

Elizabeth Thong, MBBS (UK), MRCP (UK), MMed (Internal Med)

Department of Cardiology

National University Heart Centre Singapore

1E Kent Ridge Road, NUHS Tower Block Level 9, Singapore 119228

Tel: +65 6779 5555         

E-mail: Elizabeth.thong@mohh.com.sg

Round 2

Reviewer 2 Report

The revised manuscript can be published as soon as possible.